# A human-machine interface for automatic exploration of chemical reaction networks

Miguel Steiner [1,2] & Markus Reiher [1,2] ✉

Autonomous reaction network exploration algorithms offer a systematic approach to explore mechanisms of complex chemical processes. However, the resulting reaction networks are so vast that an exploration of all potentially accessible intermediates is computationally too demanding. This renders brute-force explorations unfeasible, while explorations with completely pre-defined intermediates or hard-wired chemical constraints, such as element-specific coordination numbers, are not flexible enough for complex chemical systems. Here, we introduce a STEERING WHEEL to guide an otherwise unbiased automated exploration. The STEERING WHEEL algorithm is intuitive, generally applicable, and enables one to focus on specific regions of an emerging network. It also allows for guiding automated data generation in the context of mechanism exploration, catalyst design, and other chemical optimization challenges. The algorithm is demonstrated for reaction mechanism elucidation of transition metal catalysts. We highlight how to explore catalytic cycles in a systematic and reproducible way. The exploration objectives are fully adjustable, allowing one to harness the STEERING WHEEL for both structure-specific (accurate) calculations as well as for broad high-throughput screening of possible reaction intermediates.

An exhaustive exploration of mechanisms of chemical processes requires the automated generation of chemical reaction networks (CRNs)[1–9]. CRNs typically map chemical reactions into a graph of compound and reaction nodes[10–12]. This graph can be constructed based on automated calculations that locate transition states of reactions assumed to take place, for which various strategies exist[13–42].

First-principles investigations of reaction intermediates and transition states provide valuable insights into reaction mechanisms, as demonstrated, for instance, by numerous studies in the field of catalysis[43–57]. However, no universal, efficient, and reliable theoretical approach toward computational catalysis with generally applicable algorithms is available so that the study of a catalytic reaction mechanism of a single catalyst can require considerable time and expertise. Understanding catalysis in terms of CRNs can be a starting point for the design of cheaper, greener, and more selective catalysts[58,59], because automated procedures can analyze orders of magnitude more structures than manual approaches, leading to a far

more complete understanding of relevant reaction steps (including side and decomposition reactions) and conformations. This results in a more accurate formalization of catalytic processes and in silico predictions that cover the whole spectrum of catalyst and substrate reactivity.

The increased number of structures leads, however, to a combinatorial explosion in a brute-force exploration of all possible reactive site combinations, which prohibits the exhaustive exploration the reactivity of even moderately sized structures[5]. Based on their determination of reactive sites, automated exploration programs can be classified into two main categories. On the one hand, there are fully automated approaches[16,31,33,39,60,61] that are feasible for complex chemical systems only, if they rely on either a restrictive reactive site logic and/or computationally inexpensive calculations[5]. These conditions can, however, limit their applicability and accuracy for a particular system of interest; transition metal complexes are good examples owing to their variability in valency and generally intricate electronic

[1]ETH Zurich, Department of Chemistry and Applied Biosciences, Vladimir-Prelog-Weg 2, 8093 Zurich, Switzerland. [2]ETH Zurich, NCCR Catalysis, Vladimir-Prelog-Weg 2, 8093 Zurich, Switzerland. ✉e-mail: mreiher@ethz.ch

structures. On the other hand, a class of approaches[22,37,62,63] requires a manual setup of reactivity trials through an algorithmic interface, which can save time compared to individual structure and calculation setup, although it still relies on human decision making to determine the reactive site logic and lacks general applicability and scalability.

However, to carry out mechanism elucidations routinely, catalyst design, and other chemical optimization challenges, acceleration protocols are needed that do not corrupt any key feature of an otherwise autonomous reaction mechanism exploration algorithm. For instance, one does not want to limit mechanistic studies by pre-selecting all reaction intermediates, which brings inherent biases and constraints.

Here, we present a new algorithm driving our automated first-principles exploration approach[17,39,60,64] that allows for intuitive on-the-fly interference of an operator with an otherwise autonomous exploration, which we denote as the STEERING WHEEL. Our algorithm is able to cover all ground-state molecular compound and reaction space and can explore a CRN either in a depth-first or in a breadth-first fashion. By virtue of an integration into a graphical user interface the steering of a running exploration is straightforward and intuitive.

In the following sections, we outline the general concept of the STEERING WHEEL, discuss its implementation and its integration into our graphical interface HERON[65]. Afterwards, we demonstrate functionality and efficiency by application to several well-studied reactive systems from transition metal catalysis.

## Results and Discussion
### Conceptual design and implementation of the steering wheel
Within our modular program package SCINE[66], we have developed the automated exploration software CHEMOTON[17,39,60,64], which allows one to explore chemical reaction space based on the first principles of quantum mechanics in a single-ended manner without being constrained to specific compound or reaction types. This is achieved by defining local sites in molecular structures that are reacted with one another by pushing/pulling these potentially reactive sites together/apart and then locating a transition state. Compared to traditional (typically double-ended) transition state search algorithms, which aim at a single reaction step, our approach launches an exhaustive search for elementary steps which make no assumption on potential products. This is achieved by batch-wise writing instructions for multiple reaction trials into a database, which are then executed by processes on high-performance computing infrastructure or in the cloud[67,68]. The results of the calculations are then written back to the database and aggregated and sorted by Chemoton to construct the emerging reaction network that can then be subjected to kinetic modeling. Kinetic modeling can even be exploited for taming the combinatorial explosion of reactive events[12,69]. The number of reactive sites may also be controlled by various heuristic rules, such as first-principles heuristics that exploit properties of the wavefunction or electron density[17,70,71], graph-based rules in combination with known reactivity[72], or electronegativity-based polarization rules, where, for example, hydrogen is considered active when bound to oxygen[60,69].

However, all these approaches to restrict the combinatorial explosion of potentially reactive events are either not directed or not coarse-grained to a degree that would allow for a quick tour to potentially relevant intermediates of a reactive system. Therefore, we propose the STEERING WHEEL to allow for efficient interactive control of an otherwise autonomous mechanism exploration. Its execution is linear and the automated exploration is split into sequential exploration steps based on an on-the-fly constructed steering protocol. In a complex system, one may want to change the reactive-site determination rules based on the actual state of an exploration to assemble a flexible steering protocol that establishes key parts of a CRN first (before the exploration can dive deeper into the reactive propensity of the system). The heuristic rules can be selected from several existing

rules mentioned above (one or more of which can be based on machine learning, first principles, or graph-based rules). To enable such a workflow, we base the STEERING WHEEL on shell-like explorations. Each shell is a procedure to grow a CRN. That is, the STEERING WHEEL sets up and runs new calculations, waits for all of them to finish, and then classifies the results before further exploration steps are initiated. Reactions are, however, not limited to a specific shell, but later-found compounds can still react with the starting compounds.

The steering protocol therefore consists of two alternating exploration steps: `Network Expansion Step` and `Selection Step`. A `Network Expansion Step` is defined as an exploration step that adds new calculations and their results, i.e., structures, compounds, flasks, elementary steps, and reactions, to a growing CRN. `Selection Step` is defined as an exploration step that chooses a subset of structures (or compounds) and corresponding reactive sites from the reaction network, which limits the explored chemical space and avoids a combinatorial explosion in the subsequent expansion. For both, `Network Expansion Step` and `Selection Step`, we have developed implementations discussed in section 3.3 below. From these implementations, the operator can build the steering protocol in such a way that the desired chemical space is covered, as illustrated in Fig. 1. This steering protocol is assembled in terms of keywords – such as 'Dissociation' to initiate the search for specific dissociation reactions – by a human operator on the fly. This protocol therefore supports easy processing and may easily be generated from written form into spoken language (cf.[73–78]).

To ensure broad applicability across chemical space, the individual steps are defined in a general way, although they can be fine-tuned for each reactive system. For example, a 'Dissociation' expansion step is rather general in its definition: only dissociative reaction coordinates within a single compound are probed, but applying the step on a previous selection step can reduce the number of calculations set up from millions to hundreds or dozens. This high specificity can be achieved by combining multiple selection steps into one step, as shown for step three in Fig. 1, or by defining additional compound filters and reactive site filters – concepts available in CHEMOTON[39]: Because CHEMOTON considers a priori every structure in a network as reactive with each of its atoms as a potentially reactive site, a huge number of possible reactions arises from the combinatorial explosion of reactive atom pairings. Filters reduce the number of potential

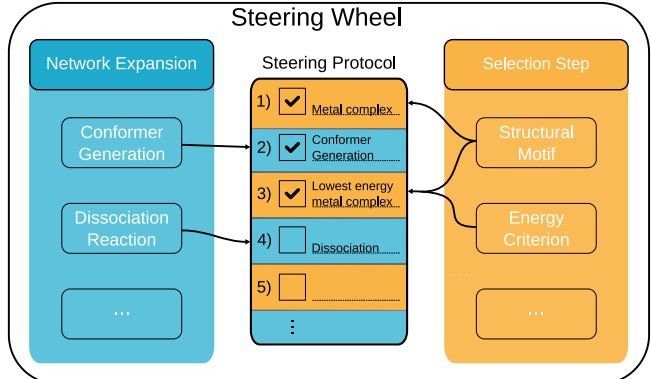

**Fig. 1 | The steering protocol (the center of the figure) is built by steps that describe network expansion and selection in an alternating fashion.** `Network Expansion Steps` (left, cyan) describe actions that add new information to the chemical reaction network (CRN); examples are "Conformer Generation" and "Dissociation", which probe all previously selected parts of the network for new conformers and dissociation reactions. `Selection Steps` (right, orange) are criteria that limit the CRN to a specific subset of compounds, structures, and reactive sites. These criteria can be based on the chemical structure ('Structural Motif') or on energy cutoffs ('Energy Criterion'), e.g., only the $n$ lowest energy conformers or compounds accessible with a given activation energy are selected.

reactions by eliminating certain structures or reactive-atom combinations from the search space. Similar to a `Selection Step`, the filters can be based on various rationales, such as graph rules or properties derived from first principles. An example for a compound filter is the `Catalyst Filter`, which allows one to define a combination of chemical elements as a catalyst and then carry out reaction trials that involve only this catalyst.

The explicit protocol for starting an exploration is not fixed, but it will evolve sequentially. The reason for this dynamic nature of the STEERING WHEEL protocol is that it cannot be known from the start what structures and reactions will be discovered, which then determines what next steps are to be enhanced or handled more restrictive. In this interactive rolling procedure, the current exploration status must be easily understandable and the potential effect of planned steps on the exploration must be foreseeable by an operator. To facilitate this immediate grasp of operator interference, the STEERING WHEEL can be executed concurrently in a Python environment and is integrated into our graphical user interface SCINE HERON[65]. The integration into HERON allows one to build exploration steps directly in the graphical user interface and then carry out the steered exploration in an intuitive problem-focused fashion. The graphical user interface displays how a potential next `Network Expansion Step` would affect the exploration by presenting the number of calculations set up for the expansion, alongside with the constructed reactive complexes and their reactive sites. Together with the existing average runtime information available in HERON, the computing time for the step can be estimated. This enables one to refine the chosen selection step to be more inclusive or exclusive based on the targeted chemical space and available resources. One such example of an expansion preview alongside the protocol is shown in Fig. 2.

While such intuitive interactions with a running exploration allow for flexible workflows, they harbor the danger of producing non-reproducible mechanism exploration campaigns. Every set of generated calculations depends on the existing results in the network and if only a random subset of calculations in the previous step were finished, it would render the exploration irreproducible. Therefore, we designed our framework in such a way that it ensures reproducibility by requiring every step of the created exploration protocol to be completed, i.e., every calculation set up must be finished, before any further manipulations of the network are permitted. The linear protocol might lead one to believe that `Network Expansion Steps` taken early in the exploration impose strong constraints on the remaining exploration. However, any `Expansion Step` can be applied on the whole CRN at any point in the protocol, meaning that any part of an explored mechanism can be studied in more detail later on with additional calculations.

The linear protocol evolution of the network is advantageous, because it allows the exploration to be completely reproducible, given the steering protocol is published. Naturally, this also requires the same versions of the applied electronic structure programs and SCINE software stack, which can be ensured by containerization that we support out-of-the-box for Apptainer[79,80]. Therefore, all explorations presented in this study are easily reproducible with the provided container and protocols deposited on Zenodo[81], where also the resulting calculations (in total 76,000 reaction trials yielding 78,000 chemical structures) are stored in a MongoDB framework[82].

In the following sections, we demonstrate how our steered automated exploration approach can be applied to study various catalytic systems. We selected three homogeneous transition-metal catalyst, which have been studied for several decades, and one heterogeneous single-site catalyst, which was recently explored with a different automated approach. A complete literature review and discussion will be impossible to achieve in the context of this work. Instead, the main focus of this work is on how the STEERING WHEEL can be applied to study complex reaction mechanisms. Although we do not achieve sufficient accuracy (because of the limited accuracy of the DFT models employed) to revoke or confirm any existing mechanistic hypothesis, each exploration includes some aspects of the mechanism where our automated search produced new results that have not been considered before.

## Propylene hydrogenation by Wilkinson catalyst
We first apply the STEERING WHEEL to the reduction of propylene by the well-known transition metal catalyst $[RhCl(PPh_3)_3]$, typically

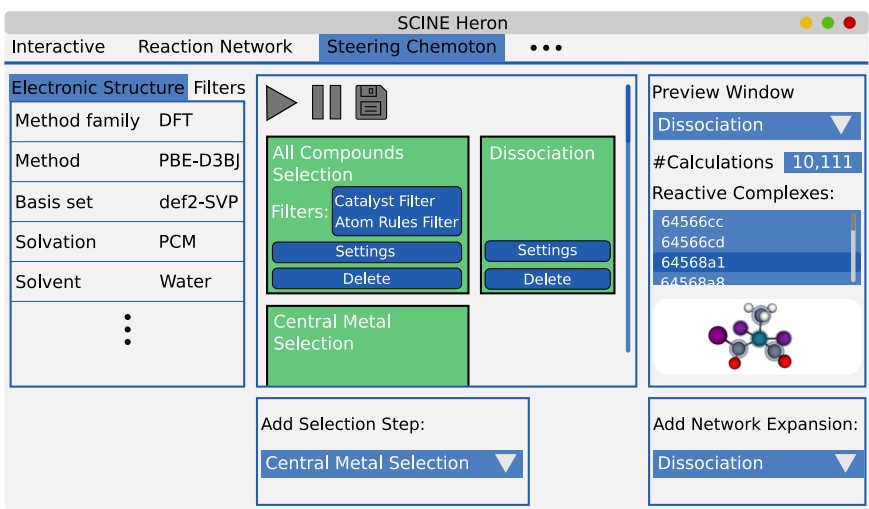

**Fig. 2 | A schematic representation of the STEERING WHEEL interface in HERON.** The tabs on the left allow one to select the electronic structure model, build specialized filters to specialize selection steps, and add user-defined settings. In the center, the current exploration protocol is displayed. The green background color signals a successful execution. The right console allows one to query the latest `Selection Step` for a potential next `Network Expansion Step`. In this case, a `Dissociation`, an abbreviation for a dissociation reaction, was selected as a potential next expansion step, meaning that the selected subset of the reaction network is probed for dissociation reactions. All the resulting calculations that would be set up for this purpose are then displayed within that console. Each potential reactive complex can be selected to be visualized as a three-dimensional structure. The blue transparent spheres in this structure represent the reactive sites of the specific structure. The pull-down menus on the bottom then allow one to add the next exploration step to the steering protocol. A screenshot of the interface in HERON is included in the Supporting Information in Supplementary Fig. 1.

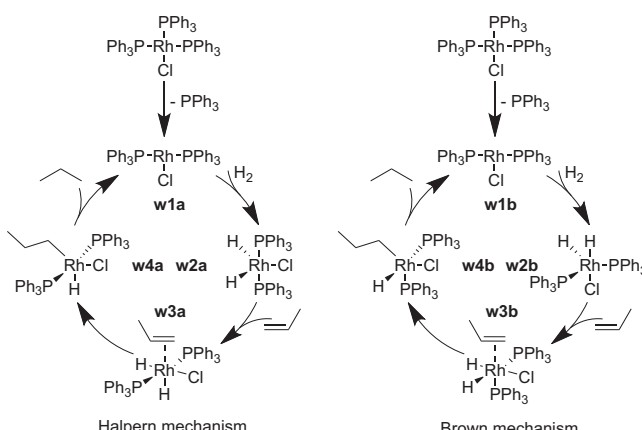

**Fig. 3 | Halpern[84] (left) and Brown[85] (right) mechanisms of the hydrogenation of propylene catalyzed by Wilkinson's catalyst.** The two mechanisms diverge with intermediate **w2a** / **w2b** (phosphine ligands in trans-position in the Halpern mechanism and in cis-position in the Brown mechanism).

referred to as Wilkinson's catalyst[83]. The two most-widely accepted mechanisms of this catalytic reaction are the Halpern mechanism[84] and the Brown mechanism[85], which are shown in Fig. 3. Both mechanisms involve catalyst activation by ligand dissociation, oxidative addition of $H_2$, olefin coordination, olefin insertion into the metal hydride bond, and reductive elimination of the alkane.

Despite the long history of research on the mechanism of this catalyst[86–89], not all intermediates of the proposed mechanism have been observed experimentally yet. The Halpern[84] and Brown mechanisms[85] diverge at the intermediate **w2**, which shows the phosphine ligands in trans-position (**w2a**) in the Halpern mechanism and in cis-position (**w2b**) in the Brown mechanism. They then differ in their rate-determining step, which is the olefin insertion in the Halpern and the product elimination in the Brown mechanism. For further details, we refer to ref. 89 and references cited therein.

Staub et al.[90] have recently explored the hydrogenation of ethylene by a simplified model of the catalyst with $PH_3$ ligands in an automated fashion based on the Artificial Force Induced Reaction (AFIR) approach[34]. Their most favored mechanism included an initial olefin insertion reaction prior to ligand dissociation and subsequent dihydrogen association. This finding is in disagreement with experimental findings[91] and can most likely be attributed to the simplification of the triphenyl phosphine ligand as $PH_3$ ligands, which has led to inconclusive theoretical results in the past[92] and was shown to be relevant for the evaluation of different accessible isomers and the energetically most favorable path[93]. We therefore included the full triphenylphosphine ligands in our exploration. Since this increased the computational cost, we limited the explored chemical space strictly to the two literature mechanisms. The steering protocol, which guided the exploration by CHEMOTON and which has been deposited on Zenodo[81], reads [`File_Input_Selection`, `Simple_Optimization`, `Central_Metal_Selection`, `Dissociation`, `All_Compounds_Selection`, `Association`, `All_Compounds_Selection`, `Association`, `Products_Selection`, `Rearrangement`, `Products_Selection`, `Rearrangement`, `Products_Selection`, `Dissociation`]. By omitting the `Selection Steps` and the initial structure optimization, the list of `Network Expansion Steps` reads [`Dissociation`, `Association`, `Association`, `Rearrangement`, `Rearrangement`, `Dissociation`]. The clear connection of our algorithmic interface and the literature mechanism is already apparent based on the strong alignment of the reaction types and the schemes in Fig. 3. The only difference between the mechanism in Fig. 3 and our steering protocol is the split of the product elimination into a `Rearrangement` and `Dissociation`,

because the formed propane was still weakly bound to the catalyst. This weak coordination is due to the semi-classical dispersion corrections, which favor non-covalent bonding in isolated species where no explicit solvent molecules stabilize the dissociated products. Even though our protocol was strictly based on the standard literature mechanism without any focus on finding diverging reaction intermediates, our selection steps and automated reaction search methods were able to find numerous isomers of the intermediates of the Halpern and Brown mechanisms during the exploration, some of which are displayed in Fig. 4. If specific isomers of intermediates are of interest or expected ones are still missing in the network, they can be searched for in a targeted manner, possibly with more accurate electronic structure methods should the electronic structure model be considered insufficient to localize them.

Already the first intermediate, the activated catalyst after ligand dissociation, features two possible isomers; that is, planar T-shaped conformations with either a phosphine ligand or the chlorine ligand in trans position to the vacant site. These isomers are known to interconvert via a trigonal planar structure[85]. Moreover, we found for many intermediates in our reaction network that the expected vacant site is partially occupied by a weakly bound hydrogen atom of one of the phenyl groups, stabilising the conformation. This agnostic interaction originates from the attractive semi-classical dispersion correction in our electronic structure description. Its relevance is difficult to assess in the present structural model due to the lack of other potential bonding partners, such as solvent molecules. The agostic bond is most pronounced in the intermediates **w2** resulting from dihydrogen association, which we found as a single concerted reaction step of hydrogen association with simultaneous breaking of the dihydrogen bond. For intermediates **w2**, all possible variations of the five-fold coordinated complex were shown to be accessible in NMR experiments of Brown et al.[85]. This observation increases the complexity of mechanisms significantly due to the numerous possible combinations of reactive intermediates. However, with our approach, we were able to find all possible isomers and variants in the ligand sphere (**w2c** - **w2h**) at once.

The agostic bond between one hydrogen atom of a phenyl group and the rhodium central ion distorts the ligand sphere from a five-fold geometry to an octahedral complex, which is most likely caused by the the semi-classical dispersion corrections in GFN2-xTB on which we relied for this exploration. We have carried out structure refinements by optimizing some minimum structures with more reliable density functional theory (DFT) methods. This converted the octahedral complex to a five-fold coordination structure as expected. The agostic interaction remained intact only in intermediates **w1c** and **w1d** due to the strong under-coordination at the rhodium ion. However, the weak hydrogen-rhodium bond did not hinder the exploration progress to find the catalytic cycle with the GFN2-xTB method. The bonding of propylene leading to intermediates **w3a**–**w3d** was possible and replaced the rhodium-hydrogen bond.

Also for intermediate **w3** we found cis- and trans-isomers. The only two possible intermediate configurations that were not found in our exploration were the two cis-phosphine conformers with either H or Cl in trans position to the $\eta^2$-bound propylene. This can be attributed to steric hindrance, because it requires the three largest ligands (i.e., the two phosphine ligands and the olefin) to be all in cis-orientation to one another, which is unlikely to be energetically favorable. We manually constructed one such isomer and optimized its structure to investigate whether it would be stable for the electronic structure model employed in the exploration. Upon structure optimization, the propylene ligand is moved further away from the ruthenium ion, featuring an elongated and weak bond between the terminal propylene carbon atom and the rhodium central ion (Mayer bond order[94] of 0.22 and bond length of 2.6 Å). Both the Mayer bond order and length exceed the detection thresholds for a stable bond in

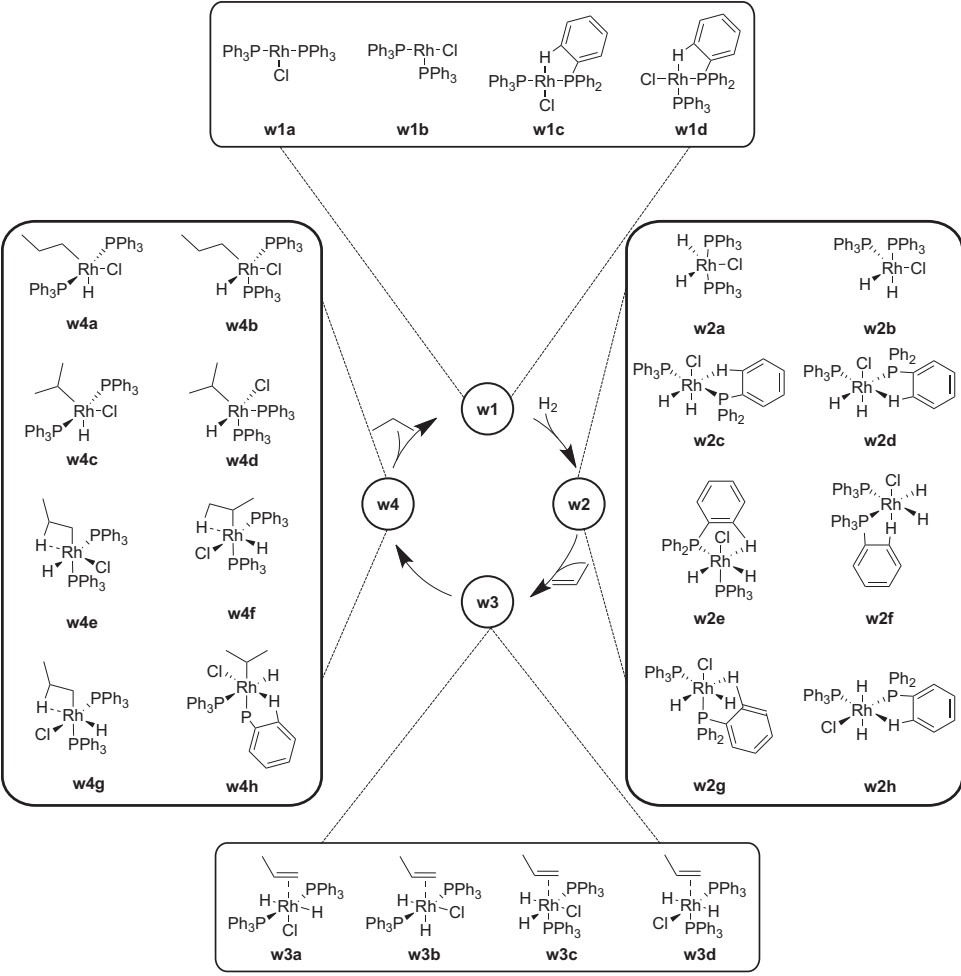

**Fig. 4 | Scheme of the on-cycle intermediates discussed in the literature and all additional intermediates that were found during exploration.** The reaction intermediates **a** belong to the Halpern mechanism[84] and intermediates **b** to the Brown mechanism[85] with the exception of **w1a**, which is part of both mechanisms.

our framework, which is why the automated reaction trials have not considered this to be a successful association reaction. Given that a full association of the propylene molecule is thermodynamically disfavored, as shown by the optimization, and that this potential reaction competes with association reactions leading to the other, energetically favored stereoisomers, we deem this unsuccessful reaction trial as correct and have not carried out further reaction explorations starting at this stereoisomer.

For the last reaction intermediate, the bound alkyl complex prior to reductive elimination, our steered exploration discovered not only the proposed terminally bound alkyl group **w4a**[89], but also intermediates with a 2-propyl ligand, for which we also found the trans- (**w4c**) and cis-phosphine (**w4d**) isomers. Furthermore, CHEMOTON located the isomers **w4e** to **w4h**, which again incorporated the weakly coordinating hydrogen atom. This hydrogen atom can either originate from phenyl or alkyl groups.

As a side remark, we note that the focus in the aforementioned work of Staub et al.[90] on the Wilkinson catalyst was on the training of a neural network potential based on the explored structures in the reaction network. This is a promising route for automated exploration algorithms, which depend on fast electronic structure methods. We chose the density functional tight-binding model GFN2-xTB, which, however, may suffer from inaccuracies in energies and sometimes also structures. The former can be easily corrected by DFT single-point calculations, whereas the latter are difficult to correct as they may lead to wrong intermediate and transition state structures and even to a

wrong topology of the emerging CRN. By contrast, system-specific neural network potentials are almost as fast to evaluate as classical force fields, but achieve the accuracy of the reference data (typically DFT)[95–104]. However, to achieve this accuracy, a huge number of reference data point (i.e., DFT single points) is required, which introduces a significant overhead before an exploration can start. Moreover, visiting new structures during the exploration may show the limitations of a parametrized neural network potential as its accuracy may deteriorate for them. These issues may be tackled by generalized neural network potentials[105–109], but one needs to be prepared for no generalist simple model achieving sufficient overall accuracy close to that of DFT. For this reason, we proposed a different scheme, called life-long machine learning potentials that can adjust in an exploration in a system-focused fashion[110].

## Ziegler–Natta propylene polymerization
Multiple polymerization reaction steps are a challenge for automated explorations due to the required number of exploration steps required to reach long-chained polymers. Therefore, as a second example, we present STEERING WHEEL results for the polymerization of propylene catalyzed by a Ziegler–Natta zirconium catalyst. The catalytic polymerization reaction is shown in Fig. 5. After activation of the stable catalyst to an active, cationic form[111], possibly facilitated by a co-catalyst, not shown in the figure and also not included in the reaction network, the polymerization is a two-step process. The to-be-inserted olefin monomer binds to a vacant coordination site of the catalyst by $\eta^2$

coordination, while the existing polymer chain is covalently bound to the zirconium in intermediate **z1**. The monomer is then inserted in a single step at the zirconium site, which generates a vacant site in intermediate **z2**. This vacant site is only weakly coordinated by an agostic C-H bond from the $\beta$ position in the polymer chain. However, this site is still available for coordination of the next monomer and does not block the polymerization. For more details, we refer to ref. 112 and references cited therein. The agostic bond increases the probability of the most common polymerization termination reaction for Ziegler–Natta-type catalysts[112], also shown in Fig. 5. The catalyst is inactivated by $\beta$-hydride transfer, in which compound **z3** is formed, and concerted release of the polymer chain with a terminal carbon double bond.

Since the termination reaction can occur in each polymerization cycle, the resulting polymerization products are of varying length. The lengths distribution can be narrowed by designing a catalyst such that the termination reaction is unfavored and only induced by the addition of a termination reagent. An automated reaction exploration can aid such catalyst design challenges as it allows one to identify rather easily all possible reaction products and study varying catalyst degradation reactions at multiple stages of the polymerization. Besides modulating the termination process, Ziegler–Natta-type catalysts allow for an elaborate ligand design to improve the stereoselectivity of the propylene insertion and, hence, to control the tacticity of the produced polymer[113–115], apart from general activity improvements based on the co-catalyst or solvent[116–120].

However, we focus on the distribution of the polymerization products and the termination reaction. We explored the polymerization with a short steering protocol with two addition reactions of the propylene monomer to the activated catalyst $[Zr(Cp)_2CH_3]^+$. The

**Fig. 5 | Cossee–Arlman mechanism[124–126] for the polymerization of propylene by a Ziegler–Natta catalyst and a possible degradation reaction via $\beta$-hydride elimination.** The two ligands abbreviated by Cp are plain cyclopentadienyl ligands without additional functional groups (hence, no enantiomeric excess in the polymerization product can be expected).

Network Expansion Steps of our steering protocol read [Association, Rearrangement, Association, Conformer_Creation, Rearrangement, Rearrangement, Dissociation]. The catalytic polymerization cycle in Fig. 5 mapped well to a simple Association, Rearrangement protocol. However, we split up the second polymerization cycle with an intermittent conformer creation step, as we noticed during the exploration that the sampling of different conformers is required in later stages of the polymerization due to the increased number of degrees of freedom in the polymer chain. We then sampled the termination reaction with these Network Expansion Steps: Rearrangement and Dissociation. The additional Dissociation step was required, because the formed product was still weakly coordinated to the catalyst, again due to the attractive semi-classical dispersion correction in GFN2-xTB and the lack of explicit solvent molecules that could replace the product by coordinating to the zirconium central ion.

We analyzed the reaction network explored in terms of the products obtained and extracted the three-dimensional structure of all compounds that do not contain zirconium. Then, we inferred the Lewis structure of each compound with XYZ2MOL[121,122] and RDKIT[123]; the result is shown in Fig. 6. After the addition reaction of propylene to $[Zr(Cp)_2CH_3]^+$ and termination by $\beta$-hydride elimination, the expected main products are 2-methyl-propene (single addition reaction) and 2,4-methyl-pentene (two addition reactions of propylene). Both were found in the reaction network and the shortest paths for their creation determined by SCINE PATHFINDER[12] were identical to the paths established in the literature[124–126]. Additionally, 18 hydrocarbon side products were found by CHEMOTON, which are shown in Fig. 6 together with the reactant propylene and the two expected products. However, their occurrence and distribution can only be considered a qualitative result, because we did not refine the GFN2-xTB reaction network with a more reliable electronic structure model such as DFT. However, the broad variety of the explored compound space highlights the capabilities of our STEERING WHEEL approach to broadly cover reaction space adjacent to that of the catalytic cycle while keeping the exploration direction aligned with the elucidation of the catalytic mechanism in question.

## Monsanto process for carbonylation of methanol

As a third example presenting a challenge for automated reaction mechanism exploration, we selected a process that involves two intertwined catalytic cycles, the production of acetic acid from methanol and carbon monoxide catalyzed by a rhodium catalyst, typically referred to as Monsanto process. The carbonylation takes place via multiple activated iodide species that are formed in solution from hydrogen iodide and are regenerated by hydrolysis of the acid

**Fig. 6 | Lewis structures of all hydrocarbons found in the reaction network starting from propylene and the zirconocene catalyst $[Zr(Cp)_2CH_3]^+$ with only two allowed addition reactions of a propylene molecule with an** alkyl-bearing Zr-catalyst. The molecular charge $q$ and spin multiplicity $M_s$ of the compounds are given below each structure ($M_s = 1$ for singlet states, $M_s = 2$ for doublet states).

iodide, which simultaneously forms the desired product. Following previous work[127–129], the intertwined catalytic cycles are depicted in Fig. 7. The reaction mechanism involves multiple insertion reactions at the transition metal complex and multiple substitution reactions without the presence of the catalyst.

The variety of compounds involved in other (non-Rh-catalyzed) reactions imposes a challenge for existing reaction filters of automated approaches, as outlined in section 2.1. The reason for this challenge is that a set of graph-based rules that define which compounds are reactive commonly activate either the organometallic (outer cycle) or solution-phase (inner cycle) reactions. A set of rules that enables the reaction exploration for both types of reactions during the whole exploration process is, however, prone to cause a combinatorial explosion due to the large chemical space spanned by such a super-set. In combination with the many subsequent reaction steps, this prohibits an exploration of the full catalytic cycle with unsupervised automated, i.e., fully autonomous, explorations. However, the reaction mechanism is also difficult to elucidate with semi-supervised explorations that rely on the specification of individual intermediates, due to multiple possible routes, stereoisomers, and bonding patterns. Because, one is interested only in the overlap of the two reaction spaces to explore the Monsanto process, our steered approach that can switch the focus of the exploration on the fly can tackle such mechanisms. By virtue of our STEERING WHEEL the required number of exploration steps and exploration flexibility is achieved easily and the intertwined catalytic cycles were found starting from methanol, hydrogen iodide (HI), carbon monoxide (CO), and [RhI₂(CO)₂]⁻ (**m1**) only.

The `Network Expansion Steps` required for the exploration of the Monsanto process were [`Association, Association, Rearrangement, Rearrangement, Association, Rearrangement, Association`], which again closely resembles the literature mechanism[127–129] shown in Fig. 7 with the only difference in the association reaction of methyl iodide to the catalyst **m1** forming intermediate **m2**. This reaction was formulated as a single elementary step in Fig. 7, but required two steps in the steering protocol described by an `Association` and `Rearrangement`.

After exploration of the reaction network with the semi-empirical electronic structure model GFN2-xTB, we refined the reaction network by carrying out DFT single-point calculations for all minimum structures and transition states (see section 3.1 for details on the computational methodology). Such a refinement is an efficient approach to improve on the accuracy of the activation and reaction energies in a CRN. Applying DFT as the next more accurate electronic structure approach is the first step of a series of available refinement approaches of increasing accuracy in SCINE CHEMOTON (such a sequence of increasingly accurate, but also more costly and hence fewer ab initio calculations can be exploited in Bayesian approaches for systematic uncertainty quantification[130,131]).

Based on the DFT activation energies and the chosen starting compounds, we searched the network for the energetically lowest paths from methanol to acetic acid with SCINE PATHFINDER[12]. This search yielded the expected catalytic path, schematically shown in Fig. 8 (**A**), but also a stoichiometric reaction of methanol to acetic acid that consumes the catalyst by forming compound **m9** shown in Fig. 8 (**B**). The two identified paths diverge at intermediate **m5** with the association reaction of a second methanol molecule in path **B** instead of CO in path **A**. The second methanol molecule hydroxylates the catalyst, undergoes carbonylation, and forms acetic acid by a concerted elimination of the methyl and acid groups, which appears to be a path not discussed in the literature so far. Because the resulting rhodium species **m9** is lacking only a CO ligand to be transformed into compound **m5**, we searched the CRN for a path from **m9** to **m5**, which would close the cycle and, hence, also classify path **B** as catalytic. This connection was not present in the CRN after the exploration with the steering protocol described above. However, after adding another `Association Network Expansion Step` to the protocol, which

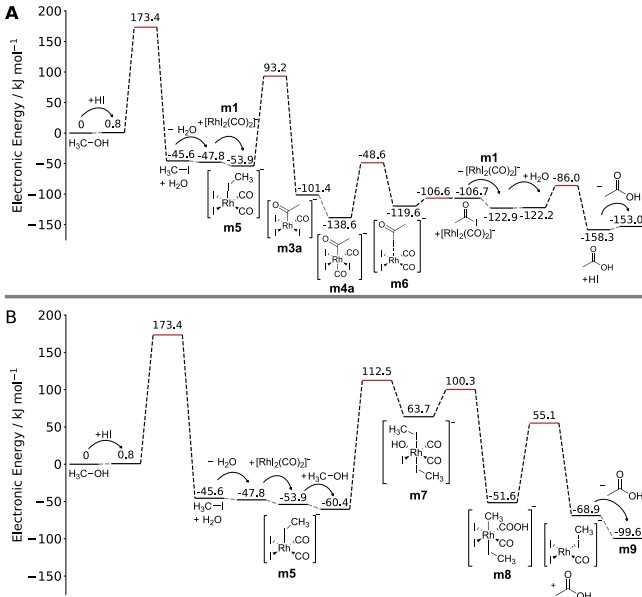

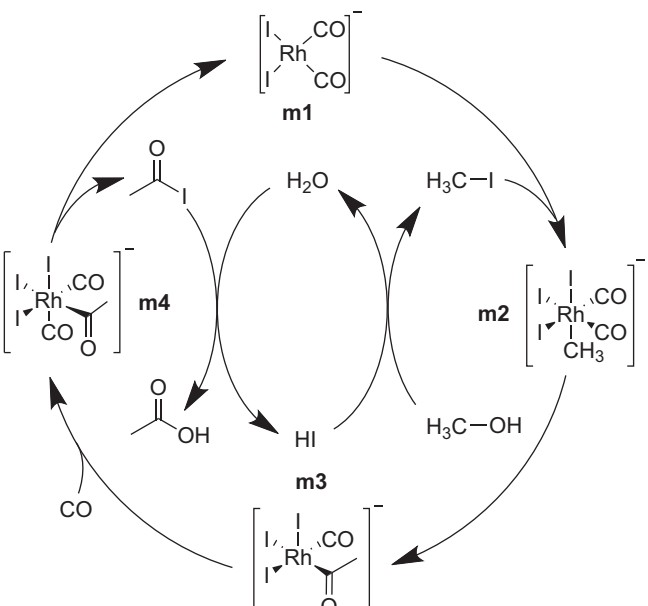

**Fig. 7 | Catalytic cycle of the Monsanto process for the carbonylation of methanol to acetic acid.** The reactants are hydrogen iodide (HI), carbon monoxide (CO), and methanol (H₃C-OH) in aqueous solution. The catalyst is the species **m1** and the product is acetic acid.

**Fig. 8 | Two competing reaction paths found in the steered exploration of the Monsanto process and identified by SCINE PATHFINDER.** Path (**A**) shows a catalytic path similar to the literature path in which the catalyst is fully recovered. Path (**B**) diverges at intermediate **m5** with the addition of a second methanol molecule instead of CO. The path still leads to acetic acid, but the catalyst requires an addition reaction of CO to recover an intermediate of the catalytic cycle. Electronic energies are based on PBE0-D3BJ/def2-TZVP single-point calculations on GFN2-xTB optimized stationary points and are given in kJ mol⁻¹. Both electronic structure methods include an implicit description of water as the solvent. Transition states are marked as red lines and barrier-less reactions are represented by gray dotted lines. The two path diagrams (**A**, **B**) were directly exported from HERON and then manually augmented with Lewis structures. Source data are provided as a Source Data file.

reacted CO with four-fold coordinated rhodium complexes that contain only a single CO ligand, we could find the missing reaction. This path is therefore an excellent example to demonstrate on how CHEMOTON can uncover new reaction mechanisms, enhanced by the intuitive reaction network analysis with PATHFINDER in HERON. Additionally, the software allows one to export a graph similar to the one depicted in Fig. 8. Fig. 8 was generated by pressing a button in the graphical user interface, which generates the level diagram, and second manually augmenting the exported SVG plot with Lewis structures and arrows. Full automatism for figure generation has not been possible, because no software exists that can generate Lewis structures of organometallic complexes reliably. However, the latter is straightforward to achieve by hand within our framework, because HERON directly provides interactive three-dimensional views of all compounds along the path.

However, the energetically lowest mechanism in our reaction network found by SCINE PATHFINDER shown in Fig. 8 (**A**) differs from the literature mechanism depicted in Fig. 7 in the sequence of carbonyl insertion and CO addition. In fact, the path shown in Fig. 8 is also not the only catalytic path we found in our exploration. All explored catalytic paths differed at the reaction step of methyl iodide addition to the planar quadratic catalyst **m1**, which we summarize in Fig. 9. The literature path involves rhodium insertion into the methyl iodide bond to form the octahedral complex **m2**, then methyl migration to form the acetyl ligand in intermediate **m3**, and subsequent CO addition to regenerate the octahedral complex **m4**[127–129]. This path was present in our reaction network and we could find multiple stereoisomers of the reaction intermediates, which differ in their ligand sphere with the methyl group binding either trans to a CO **m2a** or iodide ligand **m2**.

In addition, we found a pericyclic reaction, in which the methyliodide bond was broken and the methyl group was bound to an existing CO ligand instead of to the rhodium center, directly forming intermediate **m3a**, which is a stereoisomer of the compound **m3**, shown in Fig. 7.

The energy differences between all catalytic paths found were small and well within the uncertainty of our electronic structure description, hence further studies are necessary to discriminate the two paths. Furthermore, the activation energy of the reaction of methanol with HI was higher than expected, because this reaction is catalyzed by water[132] and we did not include explicit solvent molecules in our reaction exploration.

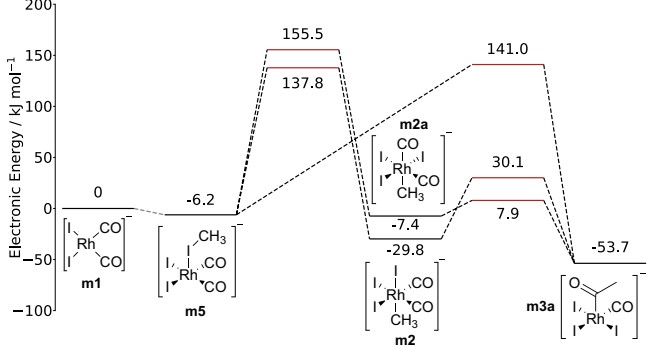

**Fig. 9 | Competing catalytic paths starting from the catalyst and methyl iodide.** The energetically lowest path found consisted of a single elementary step to break the H₃C-I bond and form the H₃C-CO bond to arrive at intermediate **m3a**. Other paths proceed via two separate transition states and differ in the ligand sphere of the reaction intermediates **m2** and **m2a**. PBE0-D3BJ/def2-TZVP single-point electronic energies were obtained for GFN2-xTB optimized stationary points (both electronic structure models with implicit water solvent, see computational methodology). Energies of transition state structures are marked by red horizontal lines. All energies are given in kJ mol⁻¹. Source data are provided as a Source Data file.

Moreover, we note that a path hypothesized in the literature[133–135], where the methyl iodide oxidative addition to rhodium is a two-step process with an initial $S_N2$-like nucleophilic attack by the rhodium complex on methyl iodide with iodide acting as a leaving group and only associating to the rhodium center in a second elementary step, could not be found by CHEMOTON in our initial exploration. Since Feliz et al. observed a strong effect of the electronic structure model and solvent description on the initial transition state[135], we suspected that this elementary step is highly unfavored in the tight binding method that we relied on for the initial structure exploration. A manual study of the elementary step also failed to locate a transition state for the nucleophilic attack. We could confirm that this failure is due to the approximate electronic structure model employed and that it is not a failure of our exploration strategy by launching another CRN exploration with the identical steering protocol but replacing the tight-binding electronic structure model with a pure DFT model (PBE-D3/def2-SVP). Furthermore, we adjusted the `Selection Step` before the third `Expansion Step` to be more restrictive, so that it considered fewer reaction trials, in order to cope with the increased computational cost per calculation. Although the algorithm carried out fewer reaction trials, it was able to locate a transition state for the nucleophilic attack of the rhodium complex on methyl iodide in the DFT-based exploration. We then added an additional `Association` step that reacted an iodide ion with five-fold coordinated rhodium complexes as this was not considered in the initial GFN2-xTB-based exploration. This step completed the $S_N2$ mechanism. The CRN of the initial steps of the Monsanto process with DFT-based reaction trials was stored in a separate database on Zenodo[81]. Hence, we could show that an existing exploration protocol can be adapted directly to another electronic structure model while allowing one to adjust the scale of the exploration depending on the costs of the electronic structure model employed.

## Silica-supported single-site catalyst

As the last challenge for the STEERING WHEEL, we selected a gallium single-site silica-supported catalyst for olefin polymerization[136]. The diverse bonding patterns in the catalytic reaction mechanism, the flexible environment of the silica support, and different possible reaction paths for various gaseous hydrocarbons that can re-adsorb to the solid-state catalyst are a challenge for automated approaches. Because of their size, periodic systems lead to calculations with high computational costs, which can prohibit extensive explorations of complex systems[5,9]. Therefore, established automated approaches in heterogeneous catalysis leverage existing literature data, group additivity, and linear scaling relations. Approaches by Goldsmith, Green, Nørskov, Reuter, Ulissi, and West have been demonstrated to be successful for pyrolysis[137–143], electrochemistry[144–147], and small molecule activation[148,149].

They delivered novel catalyst candidates and kinetic models close to experiment by incorporation of existing data[61,150,151]. High-throughput calculations have leveraged algorithms that can generate any miller index surface[152] and determine adsorbate positions[153–157]. However, the study of novel chemistry with these approaches requires either prior large scale data generation[158,159] or an extension of the incorporated reaction rules by expert developers[160].

In contrast to such data-driven approaches, there exist strategies to carry out brute-force enumerations of all species based on single-ended reaction trials without biasing the calculations to known mechanisms or energies as presented by Maeda[161–164] and Zimmerman[165,166]. However, such first-principles-based approaches require limitations in the structural model, such as exploring a single potential energy surface, constraining the nuclei of the metal surface, or restricting the studied reactions to small molecules dissociating on low-Miller-index surfaces, such as a (111) surface.

To study catalytic reactions without being constrained by existing data, exploration strategies can make a compromise between these two

extrema. On the one hand, the Liu group studied highly complex systems[167–170] by decreasing the computational costs with a machine learning potential that is tailored to the specific system based on preceding first-principles-based molecular dynamics simulations. On the other hand, the Savoie group[171] decreased the computational costs by studying a cluster model after validating it with periodic calculations, and predicting the products of each exploration shell with graph-based rules, subsequently leveraging double-ended transition state searches with constrained surface atoms, and a barrier limit to grow the CRN in a deep instead of broad manner. Hence, they showed that it is possible to study a deep CRN featuring highly complex heterogeneous species based on first-principles by restricting the search space.

Here, we show that we can reproduce and enhance their results further without any constrained atoms and solely with single-ended exploration methods by guiding the automated exploration with the STEERING WHEEL according to their mechanism hypothesis. We started our exploration with the identical cluster model of ref. 172. A gallium-ethyl species, labeled **H1** in Fig. 10, was probed for ethylene association reactions, producing species **H2** and so on. The labels up to **H17** are identical to previous work[171], all higher numbers are newly found gallium species by our protocol. The exploration required an increased number of exploration steps compared to the other CRNs due to high number of consecutive elementary steps of the mechanism. In total, our steering protocol consisted of 19 `Network Expansion Steps`.

In addition to the known reaction pathways yielding 1-butene, ethane, cis-butene, isobutylene, propylene, and polymerization (up to $C_5$- and $C_6$-species), our protocol could locate pathways to 1,3-butadiene, trans-butene, and alternative pathways to the known products. In view of the new path to trans-butene, we can dismiss the speculated[171] enantioselective preference in 2-butene production as the two reaction pathways are energetically identical within the

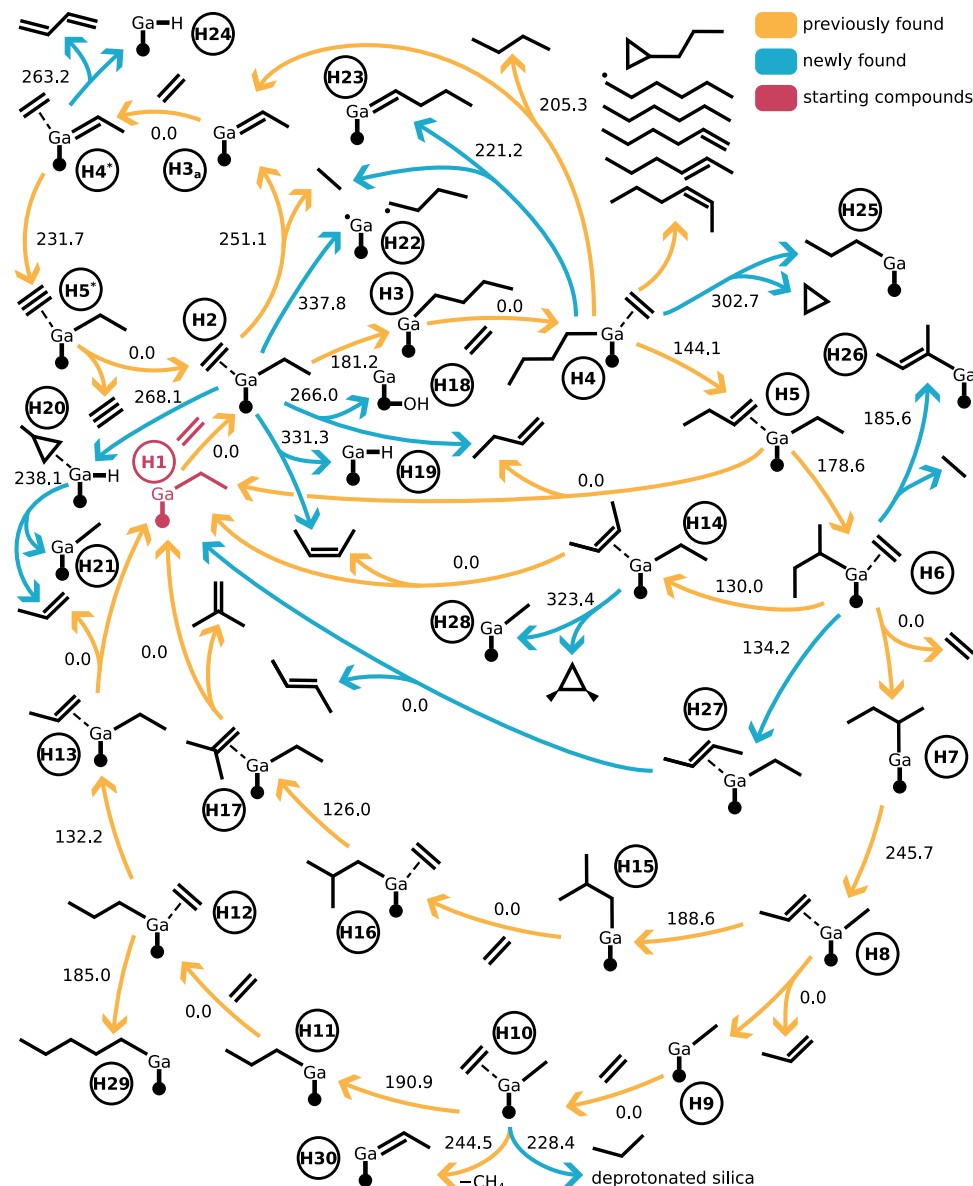

**Fig. 10 | Competing catalytic paths and degradation reactions starting from the gallium-ethyl species H1 and ethene.** The three different colors represent previously explored reactions[171] (orange), reactions newly discovered by our protocol (blue), and the two starting compounds (red). The literature-known reactions (in orange) were also found by our exploration. The bond between 'Ga' and the dot represents the multiple bonds between the gallium ion and oxygen nuclei of the silica surface. Each reaction arrow is labeled with the electronic activation energy of the reaction in kJ mol$^{-1}$. Barrierless reactions, both endo- and exothermic, are labeled with '0.0'. The activation energies are based on B3LYP-D3BJ/6-311G** single-point calculations on GFN2-xTB optimized stationary points.

uncertainty of our electronic structure model. Furthermore, the path to 1,3-butadiene agrees with other mechanistic studies[173].

Although our exploration exploited knowledge of the reaction mechanisms by Savoie[171], we stress that our approach of steered automated explorations can also be applied in the case of vague or even conflicting ideas about a reaction mechanism. Our approach works in these cases as well, because pathways close in reaction space are sampled together with intended ones, and the exploration strategy can be adapted to the failure of finding certain species or pathways. The latter occurred in this study. Ref. 171 presented the shift of the methyl group in the reaction from **H7** to **H8** as a shift of the methyl group in $\beta$-position to the gallium ion. We first failed to find reaction paths to species **H8**. Then, we studied the three-dimensional structure and were certain that species **H8** is accessible mainly by a shift of the methyl group in $\gamma$-position to the gallium ion. Therefore, we included this additional option into our steering protocol and could drive the exploration successfully to **H8** and beyond.

In terms of computational resources and extensiveness of the explored chemical space, we note that our exploration required slightly more computational time (1272 compared to roughly 900[171] days in serial computing time, cf. section 1 in the Supporting Information for the definition). However, we did not constrain any nuclei and could therefore explore more degradation reactions featuring interactions with the silica support, and also found new reaction paths by mapping out more of the chemical reaction space. The exact scale of CRNs produced with different automated approaches is difficult to compare, however, because the different approaches have varying definitions of elementary steps and structures, and different de-duplication algorithms, i.e., algorithms that identify two independently found structures to be identical.

Our CRN of the gallium single-site catalysis encompasses 1795 compounds and 1948 flasks that aggregate a total of 37,053 structures, which were connected by 14,118 elementary steps that were aggregated into 4,533 reactions (for a definition of these terms, see section 1 in the Supporting Information and ref. 4). The aggregation of structures into compounds or flasks (that is, a collection of non-covalently bound molecules) is based on identical molecular charge, spin multiplicity, and molecular graph as determined by MOLASSEMBLER[174,175]. Ref. 171 does not specify the total size of the CRN. The supporting information[176] contains in total 134 transition states, meaning that even if all of them belong to unique reactions, i.e., no two transition states connect the same set of compounds, our CRN is about ten-fold in size in terms of unique reactions (4533).

However, we also note that comparisons between automated reaction network explorations, even if applied to the same chemical system, are generally difficult and still an open challenge in this field[177], due to numerous options in many programs, varying reaction exploration strategies, different storage of the reaction network, and the challenge to compare highly complex data structures that represent chemical reaction networks.

Since we did not constrain the silica support during our exploration, we witnessed our de-duplication algorithm, which is based on a molecular graph isomorphism[174], struggle in some cases in this reaction network because it could not distinguish two compounds that differ by slight variations in the silica support. However, this is not a drawback of the algorithm, but a consequence of the actual feature of the surface, presenting a variable support for the reaction to take place.

## Conclusions

We have developed a general framework, the STEERING WHEEL, for intuitively guiding automated reaction mechanism exploration campaigns, while ensuring the creation of reproducible and transferable workflows. The design of our algorithm allows for straightforward monitoring of the exploration progress as well as for inquiring

subsequent `Network Expansion Steps`, which allows one to target wanted regions of chemical reaction space without the need to specify individual intermediate compounds or even structures. This improves the feedback on exploration decisions and facilitates the planning of subsequent exploration steps.

While the framework allows for efficient explorations based on human input, each exploration step and therefore the complete network can be pushed towards exhaustive exploration (i.e., considering any pair of atoms of any nodes of the reaction network as reactive) at any point in the workflow. This allows one to study catalytic mechanisms routinely in a rather complete fashion with minimal human work or domain knowledge in the setup of electronic structure calculations and automated explorations. We emphasize, however, that our framework is not limited to catalytic mechanisms, but can be applied to explore chemical reactivity in general.

We have applied the STEERING WHEEL to three well-known homogeneous transition metal catalysts and one heterogeneous single-site catalyst. For the Wilkinson catalyst, our exploration covered both literature mechanisms as well as additional potentially relevant reaction intermediates. The effect of the triphenyl phosphine ligands upon the optimized reaction intermediates and transition states due to their strong steric effect suggests that their explicit inclusion in the structural model of theoretical studies of this mechanism is important.

For the Ziegler–Natta metallocene catalyst, our exploration covered the literature reaction path to the expected polymerization product including the correct termination reaction. Additionally, we found other homo- and heterolytic termination reactions that allowed us to cover the reaction space in such a way to find 18 other hydrocarbon polymerization (by-)products after only two addition reactions of propylene with the catalyst.

We note that no quantitative evaluation is possible from our Wilkinson and Ziegler–Natta reaction networks as this would require a refinement of the networks with more accurate electronic structure methods (such as DFT). We have, however, carried out a DFT refinement of the reaction and activation energies of the reaction network containing the most compounds in this study, namely the exploration of the Monsanto process. In this network, we could find the intertwined catalytic cycles spanning six subsequent reactions from the reactant to the product as well as an unreported mechanism that produces acetic acid in an additional catalytic cycle initialized by a second association reaction of methanol with a pentavalent rhodium species. Additionally, we found multiple alternative paths for the reaction with the highest activation energy in the catalytic mechanism, the oxidative addition of methyl iodide to the catalyst.

The two findings, the catalyst degrading mechanism and the alternative paths in the catalytic cycle, highlight how an automated and systematic, yet guidable, algorithm allows one to study complex reaction mechanisms in great detail. However, we also note that the exploration of the Monsanto process overestimated at least one activation energy due to missing explicit solvent molecules in the exploration, which therefore lacked catalytic solvent effects, and missed one previously reported reaction path for the addition reaction of methyl iodide due to the selection of the fast GFN2-xTB model of limited accuracy for initial structure explorations, which we confirmed with a second, but limited automated exploration of the mechanism with DFT-based reaction trials. Hence, it can be advantageous to apply our STEERING WHEEL algorithm to restrict reaction network explorations in such a way to reduce the required number of calculations such that initial DFT structure explorations are feasible and introduce systematic solvation correction protocols, which are currently under development in our group.

For the single-site gallium silicate catalyst, we could push the boundaries of accessible deep reaction mechanisms by exploring a mechanism spanning twelve subsequent elementary steps. We could recover the known reaction network completely and found additional

reaction paths resulting in other (by-)products and alternative paths to already known products. We achieved this with single-ended exploration methods without explicit assumption of the products and without constrained nuclei. This demonstrates that our approach is general and applicable to a broad range of systems. It quickly maps out the relevant chemical reaction space and systematically improves on existing data and hypotheses.

The modular infrastructure of SCINE in general, SCINE CHEMOTON in particular, and of our STEERING WHEEL algorithm form a suitable basis for further extensions of individual parts of automated workflows, such as more advanced `Network Expansion Steps` (e.g., reaction trials featuring multiple electronic structure models[31], systematic network refinement with more accurate electronic structure methods[68] or with automated microsolvation approaches[178–181], or more exhaustive conformer generation[174,182–184]). Moreover, inclusion of `Selection Steps` that do not rely on human input, such as general heuristics derived from first principles[71], results from existing explorations[72], machine learning[185], path information[12], or kinetic simulations[69,186–192] is straightforward and will further enhance the capabilities of the STEERING WHEEL, which has been implemented into our graphical user interface SCINE HERON, which is available free of charge and open source.

## Methods
### Computational methodology
All data management, quantum chemical calculations, and structure manipulations were conducted within our general software framework SCINE[66], which is available open source and free of charge. The STEERING WHEEL was implemented in our graphical user interface SCINE HERON[65], with SCINE CHEMOTON[39,64] as the underlying engine to drive the mechanistic explorations. All reaction trials were carried out with the `Newton Trajectory 2` (NT2) algorithm[39,193,194], the reactive sites were determined by the selection steps made and filters chosen, which are stored within the provided protocol files deposited on Zenodo[81]. All reaction trials were carried out with the SCINE CHEMOTON default settings (also provided in the protocol files) and all calculations were carried out with a SCINE PUFFIN[67] Apptainer container. The molecular graphs required for sorting all chemical structures into compounds and flasks were constructed by SCINE MOLASSEMBLER[174,175], which also enabled the generation of conformers of the Ziegler–Natta zirconium catalyst based on distance geometry.

All electronic structure calculations were carried out by external programs, which can be controlled by the SCINE interface[195] that allows to freely select and substitute the underlying electronic-structure model, including hybrid models[196]. All explorations were initially carried out with GFN2-xTB[197] as implemented in xtb 6.5.1[198] supported by our interface[199]. Further refinement of the Monsanto network was carried out with the (pure) generalized-gradient-approximation Perdew–Burke–Ernzerhof[200,201] (PBE) exchange-correlation functional with 25 % exact exchange (PBE0)[202]. These PBE0 calculations were carried out with TURBOMOLE 7.4.1[203] with the def2-TZVP basis set[204]. The refinement of reaction and activation energies in the CRN of the gallium single-site catalyst was carried out with Becke-3–Lee–Yang–Parr (B3LYP) exchange-correlation functional[205–208] and the 6-311G** basis set[209,210] in order to compare to the network published in ref. [171]. The exploration of the first steps of the Monsanto network with DFT validates the reaction mechanism obtained with the more approximate electronic structure model. It was carried out with the PBE functional and the def2-SVP basis set[204] with ORCA 5.0.3[211]. All DFT calculations included the D3 dispersion correction[212] with Becke–Johnson damping[213] and density-fitting resolution of the identity through an auxiliary basis set[214].

The exploration of the Monsanto process was considered with an implicit solvation description applying the dielectric constant of water.

As solvation models we applied (i) the Conductor-like screening model[215] for the PBE0/def2-TZVP single-point calculations, (ii) the Gaussian charge scheme[216,217] for the exploration with PBE/def2-SVP, and (iii) the generalized Born solvation area model[218–221] for the GFN2-xTB tight-binding calculations. All calculations and their results were stored in our MongoDB-based database format[82] on Zenodo[81].

### Methodological developments
We have improved the NT2 algorithm regarding the transition state guess. Previously, the highest local maximum along a search trajectory had been selected as transition state guess, which could cause problems for atoms that get too close at one end of the search trajectory as those configurations do not represent transition states, but some arbitrary high-energy structures. We improved the selection based on the observed bond order changes during the trajectory. If the desired bond order change has occurred during the trajectory, the last local energy maximum before the event will be selected as a transition state guess. If the desired bond order change occurs, but our algorithm has not observed any local energy maximum up to this point, as determined by a screening window after smoothing the curve with a Savitzky–Golay filter[222], the first local maximum after the event will be selected. If the bond order change is not observed during the trajectory, e.g., due to a failing calculation before the required bond order threshold could be reached, the highest local energy maximum structure will be selected as a transition state guess.

Moreover, we improved the NT2 stop criteria and forces for haptic bond formations, which are crucial for many transition metal catalyzed reactions, due to possible $\eta^n$ coordination modi. The NT2 algorithm is based on the construction of a reaction coordinate based on pairs of nuclei, which in general allows it to probe more complex reaction coordinates than fragment-based approaches such as NT1[39] and AFIR[34]. However, the combination of pairs of single nuclei may lead to problematic force additions for highly complex reaction coordinates involving a haptic bond formation and another association reaction or multiple dissociation reactions involving the same reactive site multiple times. One such example is a $S_N2$-like reaction with one substituent forming a haptic bond. We have improved on the NT2 algorithm in such a way that the involvement of haptic bond formation or breaking is detected based on calculated bond orders, which allows the software to deduce whether pairs of nuclei belong to the same molecule or not. Based on this information, the applied forces on the reacting nuclei are scaled such that the intended $\eta$ bond formation or breaking is possible without eliminating other concerted reaction coordinates.

Another notable improvement to CHEMOTON's reaction exploration capabilities is the addition to carry out a fast screening of potential dissociation energies, which allows the software to skip the more expensive NT2-based algorithm. For this, the to-be-dissociated structure can be split into multiple molecules determined by CHEMOTON's reactive-site logic. The two or more separate molecules can then be optimized separately to obtain the products of the dissociation and reaction energy. The optimizations are carried out for multiple possible charge combinations, to consider that the bond(s) can be cut in a homo- or heterolytic fashion, as well as for different spin multiplicities to account for different spin distributions in the fragments. For the combination that yields the lowest dissociation energy, the software then probes the optimized fragments for a barrierless reaction by placing the fragments alongside the cut bonds with the distance elongated to the sum of van der Waals radii of the reactive sites. If the optimization of this super-system then yields the initially dissociated structure, a barrierless elementary step is added to the CRN. Hence, barrierless reactions are found with minimal computational cost. Such barrierless reactions can be crucial for catalyst activation (see e.g., the activation of Wilkinson's catalyst).

In the explorations reported in this work, all dissociation reactions with a reaction energy below 200 kJ/mol were additionally sampled

with our conventional NT2-based algorithm afterwards to verify the results from the faster algorithm.

### Implementation details of exploration steps and exploration workflows

The steered explorations are carried out by the `Steering_Wheel` Python data structure, which receives an exploration protocol as a list of exploration steps that are either a `Selection Step` or `Network Expansion Step`. The management of technical details such as individually forked processes, database information forwarding, and database querying are handled by this data structure. The implementation in HERON provides further abstractions and the operator can generally operate the steered exploration by selecting options from the existing implementations which are sufficiently general so that all explorations presented in this work can be carried out in the graphical user interface.

We have developed a number of STEERING WHEEL exploration steps that we expect to be well-suited for most molecular chemistry including homogeneous transition-metal and single-site catalysis. The implemented `Network Expansion Steps` allow one to generate conformers with SCINE MOLASSEMBLER[174,175] and to extend the CRN by steps that are encoded in basic chemical language; for example, `Association` for the association reaction of two molecules, `Dissociation` for the dissociation of a bond in a single molecule, and `Rearrangement` for rearrangements within a molecule by intramolecular reaction. A complete list of the implemented `Network Expansion Steps` is given in Supplementary Table 2 in the Supporting Information. Generally, the various `Expansion Steps` can be chosen such that either bi- or unimolecular reactions are sampled. This guides the CRN in the general direction of either aggregating reactants or progressing the reactivity of already activated compounds. Additional settings include the number of sampled reaction coordinates, e.g., a straightforward ligand association reaction can be sampled with a single reaction coordinate, while complex rearrangements of haptic bonds can involve multiple associative and dissociative coordinates.

The specificity of the `Expansion Steps` is achieved by the preceding `Selection Step`. The currently implemented `Selection Steps` allow one to continue with the products found based on different structural or energy criteria and are listed in Supplementary Table 3 in the Supporting Information. Relevant conformers can be selected based on their relative energies and/or based on maximum structural diversity in order to cover the phase space of reactants as much as possible enabled by clustering structures (e.g., according to their root mean square deviation).

Reactive sites of compounds and structures can be limited by various heuristic rules. To this set of rules, we have added a reactive site filter to carry out reaction trials only in the vicinity of a chemical element. For easy usability, this new filter was combined with a suitable compound filter in a `Selection Step`, the `Central_Metal_Selection`, that allows one to focus reaction trials strongly on a central ion and its vicinity, a concept that is particularly relevant for transition metal chemistry with the central ion orchestrating the chemical transformations. This and other `Selection Steps` may also be specialized up to the point of choosing a single pair of atoms within two specific structures as the sole reactive sites in the whole CRN with the integration of the existing filtering logic from CHEMOTON, as discussed in section 2.1, which has been enhanced with a general framework to build sets of reaction rules within HERON[65] as shown in Figure 1 in the Supporting Information. This enables one to apply different `Selection Steps` in a very flexible way. If a system can be described well by a general set of reaction rules, e.g., as is often the case in organic chemistry, the filters can be set for multiple steps in the exploration and the general reactivity is guided based on the available resources. However, if highly diverse chemical reactivity shall be explored within one CRN, such as in the Monsanto process that

involves both hydrolysis and condensation reactions of small molecules and reactions with an organometallic catalyst, frequent changes of the applied reaction rules can efficiently shift the focus of the steered exploration.

In the unlikely case that none of the current implementations is sufficient to explore a particular system, further additions to our framework are straightforward. A new aggregate filter can be generated by defining a single method that takes either one or two aggregates and specifies if these are to be considered as reactive or not. Within that method the two aggregates can also be queried for more detailed information such as their molecular graph, charge, and more. A new reactive site filter is defined by methods that take potential reaction coordinates of a given molecular structure and returns a list of valid reaction coordinates.

Additionally, new `Network Expansion` and `Selection Steps` can be implemented. The linear steering protocol is expected to consist of alternating `Network Expansion` and `Selection Steps`. Therefore, each exploration step must be able to process the output of the step before and produce an output that can serve as an input to the next one. These input and outputs are encoded in specific data structures in CHEMOTON. A `Selection Step` produces a result that specifies to-be-applied filters and / or specific individual structures. A `Network Expansion Step` produces a list of all compounds, flasks, structures, and reactions that it has modified. A new `Selection Step` is implemented by defining a method that takes the result of a `Network Expansion` and constructs the list of valid structures. A new `Network Expansion` defines the specific jobs it must execute, the different CHEMOTON gears it must execute, and lastly, how to execute them and then collect the results by a database query.

We would like to stress that most applications will not require such development work, but can directly apply the existing exploration framework by selecting from the existing implementations.

### Reporting summary

Further information on research design is available in the Nature Portfolio Reporting Summary linked to this article.

## Data availability

The reaction network data generated in this study have been deposited in the Zenodo database under accession code 10611686[81]. The networks are stored in our MongoDB framework alongside with the exploration protocols, a description on how to load the data and reproduce it with our software, and the Apptainer container of SCINE PUFFIN[67] that carried out all calculations. The additional DFT-based exploration of the first steps in the Monsanto process is also deposited as a separate database in the same repository. The Supplementary Information includes metadata of the reaction networks, such as number of compounds and required computing time, and a summary of all implementations of `Network Expansion Steps`, `Selection Steps`, and filters. Source data are provided with this paper for Figs. 8 and 9 and the tables in the Supporting Information. Source data are provided with this paper.

## Code availability

The underlying SCINE software stack as well as the new graphical user interface are freely available and open-source[66]. The STEERING WHEEL software framework within SCINE CHEMOTON has already been released in version 3.1. The explorations of Wilkinson's catalyst, Ziegler–Natta catalyst, and the Monsanto process were carried out with this version. The exploration of the gallium single-site catalyst requires the generation of reaction trials of non-covalently bound reactive complexes, which are added to SCINE CHEMOTON in version 3.2. A description on how to install a pre-release version of these features and the graphical user interface is given alongside the data archive on Zenodo[81]. In addition to the publicly available release, we

note that HERON and CHEMOTON have been included into the AutoRXN workflow[68] on Microsoft Azure and Azure Quantum Elements[223,224].

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

## Acknowledgements
This publication was created as part of NCCR Catalysis, a National Centre of Competence in Research funded by the Swiss National Science Foundation (grant number 180544). MS gratefully acknowledges a Swiss Government Excellence Scholarship for Foreign Scholars and Artists (2020.0047).

## Author contributions
The project was conceived by both authors. M.S. wrote the software and carried out the calculations. Both authors analyzed the results and prepared the manuscript. Both authors acquired funding. M.R. acquired the computing resources and supervised the project. M.R. is the corresponding author.

## Competing interests
The authors declare no competing interests.
