## [Peer Review File · Nature Communications]

A human-machine interface for automatic exploration of chemical reaction networksREVIEWER COMMENTS

Reviewer #1 (Remarks to the Author):

Automation combined with state-of-the-art machine learning techniques will greatly benefit the understanding of complex reactions and chemical processes. This work developed the STEERING WHEEL algorithm to address the mechanism exploration issue for the typical automation strategy, while three well-known metal catalysts were used to test the power of the code. Overall, it is a nice piece of software but the whole manuscript is more like a tutorial instead of a scientific paper. Meanwhile, the chosen catalysts are relatively simple, I mean, compared with the solid catalysts in heterogeneous catalysis. I was skeptical about the extension application of this code in reactions on the surfaces of solid materials, which are clearly better options for test purposes. Other well-developed codes such as RMG (Angew. Chem. Int. Ed. 2023, 62, e2023065) and Catkit (J. Phys. Chem. A 2019, 123, 2281–2285) also worked very well for automation and catalyst design.

Reviewer #2 (Remarks to the Author):

The article "Navigating chemical reaction space with a steering wheel" describes the capabilities of the STEERING WHEEL algorithm for human-guided automated exploration of complex chemical reaction networks via quantum chemical simulations. STEERING WHEEL provides an intuitive framework for human operators to define the exploration protocol and allows to decide on future steps based on the outcome of previous steps. Overall, I think the paper is well-written, describes a significant advance of the state of the art in the field and is of broad interest in chemistry. However, I think there are still multiple issues that require additional attention by the authors before publication. Hence, I recommend major revisions before this work can be published. Please find my detailed comments below.

Major issues:

One part that I think is currently missing in the main text is a brief explanation of all the possible network expansion steps for the steering protocols. The authors provide steering protocols multiple times in the text but it is unclear what they actually constitute. This is somewhat explained in section 5.3 but I think summarizing the key features of these steps in a table in the main text would be crucial. Additionally, it is not clear how the authors came up with these protocols. Only the following is mentioned in the text: "This steering protocol is assembled in terms of keywords – such as 'Dissociation' to initiate the search for specific dissociation reactions – by a human operator on the fly." So I imagine that based on the outcome of a previous step, a human operator decides on what next step to choose. I think it would be insightful to outline how the final steering protocols was arrived at and not only have the final protocol there. I understand the final protocol is important for reproducibility but I think the process of building the protocol is as important if the reader wants to apply the tool to their specific problem. This is addressed somewhat in the Ziegler-Natta case study but I think this should be elaborated on more extensively also in the propylene hydrogenation case study and the Monsanto process case study. I am particularly interested to read how a specific observation makes the reader choose one over the other step.

Similarly, I am missing a more technical description of what the implemented network expansion and selection steps actually do. In my opinion, currently, this is only described in a relatively superficial manner in the methods section of the manuscript.

The paper presents case studies showing the application of STEERING WHEEL to study catalytic mechanisms of transition metal-catalyzed reactions. What I am missing are comparisons to classical expert-guided computational explorations of the same mechanisms. For instance, here are two references that could be compared to for the first case study (there might be others): <https://doi.org/10.1021/ja00220a010>, <https://doi.org/10.1246/bcsj.20120113>

Page 11: The authors state the following: "Also for intermediate w3 we found cis- and trans-isomers. The only two possible intermediate configurations that were not found in our exploration were the two cis-phosphine conformers with either H or Cl in trans position to the η^2 -bound propylene." Subsequently, the authors speculate that these intermediates are likely not accessible. I think it would be best to simply check this by computing these intermediates by hand and confirm that they are inaccessible.

Page 12: The authors state the following: "By contrast, neural network potentials are as fast to evaluate as classical force fields, but achieve the accuracy of the reference data (typically DFT)." I think this statement needs to be supported by literature. It seems to me that the authors refer here to system-specific neural network potentials. I think this needs to be worded more clearly. This is important as I think most general neural network potentials are slower than classical force fields and largely comparable to semiempirical quantum chemistry methods. This has been discussed extensively in a recent review: <https://doi.org/10.1021/acs.chemrev.0c01111>

Page 15: The authors state the following: "The reason for this challenge is that one cannot define a set of graph-based rules that evaluate all intermediates of the reaction mechanism given in Fig. 7 as reactive while also restricting the search space in such a way that a combinatorial explosion of reaction trials is avoided." I do not understand how the authors can state that it is impossible to define a set of graph-based rules. I think the authors should elaborate on that.

Page 16: The authors state the following: "This search yielded the expected catalytic path, schematically shown in Fig. 8 (A), but also a stoichiometric reaction of methanol to acetic acid that consumes the catalyst by forming compound m9 shown in Fig. 8 (B)." What does "consumes" here mean exactly? Initially, my interpretation was that this is a catalyst deactivation process. However, looking at the structure of m9, I would expect it to be possible to react with another molecule of CO to eventually get back to intermediate m5. Did the authors identify such a pathway? If this is indeed a facile process, I would not use the word "consumes" but I would rather call it an alternative mechanistic pathway. At least thermodynamically, it does not seem to be an energy sink so I would imagine that one can get back to catalytically active intermediates.

Page 19: The authors state the following: "Since Feliz et al. observed a strong effect of the electronic structure model and solvent description on the initial transition state [120], we suspect that this elementary step is highly disfavored in the tight binding method that we relied on for the initial structure exploration." As in one of my comments above, I think it would be good to simply calculate this step explicitly to check whether the hypothesis that this elementary step is simply highly unfavorable at the level of theory used is correct.

Minor issues:

Figure 6: I think the radical dots should be made a bit thicker, I find them easy to be overlooked.

The authors state the following: "This path is therefore an excellent example to demonstrate on how Chemoton can uncover new reaction mechanisms, enhanced by the intuitive reaction network analysis with Pathfinder in Heron that allows one to export a graph as depicted in Fig. 8, simply by pressing a button in the graphical user interface." I find this statement slightly misleading. Initially, I was quite impressed with this statement. However, when reading the caption of Figure 8, the authors state: "The two path diagrams A and B were directly exported from Heron and then manually augmented with Lewis structures." Hence, I think the authors should already mention in the main text that the figure is augmented. In that regard, I wonder whether the non-Lewis structures (chemical sum formulas) were also added later or whether they are added by the program. Additionally, are the arrows and compound labels also added by hand?

This is not really an issue but more of a comment. The authors state the following: "In addition, we found a single-step process, in which the methyl-iodide bond was concertedly broken and the methyl group was bound to an existing CO ligand instead of to the rhodium center, directly forming intermediate m3a, which is a stereoisomer of the compound m3, shown in Fig. 7." Personally, I would refer to this step as a pericyclic reaction as it has a cyclic transition step and is concerted. I would even be tempted to refer to it as a cycloaddition, even though, strictly, cycloadditions involve "unsaturated molecules".

Reviewer #3 (Remarks to the Author):

Reviewer #4 (Remarks to the Author):

The authors introduced a steering wheel algorithm, which provides an intuitive means of directing the exploration of reaction networks. This algorithm incorporates both a network expansion step and a selection step, enabling the generation of conformers to expand chemical reaction networks while also constraining the explored chemical space. To illustrate its effectiveness, the authors have applied this algorithm to three widely recognized transition metal catalysts, yielding promising results. I believe that this algorithm holds the potential to efficiently automated reaction mechanisms in a more controlled way. In order to recommend publication, the following concerns need to be addressed.

1. In Section 2, while the authors attempt to present the conceptual design and implementation of the steering wheel, it is very hard to follow without prior knowledge of SCINE and CHEMOTON. The authors should provide a more detailed explanation of the connections between these elements and clarify the logical integration of the steering wheel.

2. The authors have provided three case studies. Nevertheless, in order to more effectively showcase the benefits of the steering wheel concept, it is advisable to include a fair comparison with previous approaches. This comparison could encompass factors such as the number of calculations and suggested intermediates. If only the steering wheel method proves effective in these case studies, the necessity of this approach should be underscored.

3. In these three case studies, GFN2xTB were employed. However, structure refinement through DFT was only applied on Wilkinson catalyst (first case study). Since tight binding approach may lead to wrong structure, this omission may introduce inaccuracies into the resulting reaction mechanisms and proposed intermediates for the second and third case studies. The authors should address how this choice impacts their findings.

4. In each case study, the authors have provided a list of network expansion steps, exemplified by [Dissociation, Association, Association, Rearrangement, Rearrangement, Dissociation] in the case of Wilkinson catalyst. It is my belief that the sequence in which these expansion steps are undertaken can significantly influence the resulting reaction mechanism. Therefore, the authors should clarify how one can systematically consider the order of expansion steps without making prior assumptions about the reaction mechanism, and how to handle the case that multiple steps happen simultaneously.

5. In alignment with point 4, since the definitions of expansion steps and selection steps are crucial to the steering wheel algorithm, I recommend that the authors provide a more comprehensive explanation, supplemented by illustrative examples of their design. This would not only enhance comprehension but also facilitate a better understanding of the steering wheel interface (as shown in Figure 2).

List of Changes for Manuscript NCOMMS-23-40908

“Navigating chemical reaction space with a steering wheel”

Miguel Steiner and Markus Reiher

We thank the reviewers for their critical reading of the paper and for the suggestions on how to improve on it. In the following, we address the issues raised by the reviewers point by point.

Comments of Reviewer 1

Automation combined with state-of-the-art machine learning techniques will greatly benefit the understanding of complex reactions and chemical processes. This work developed the STEERING WHEEL algorithm to address the mechanism exploration issue for the typical automation strategy, while three well-known metal catalysts were used to test the power of the code. Overall, it is a nice piece of software but the whole manuscript is more like a tutorial instead of a scientific paper. Meanwhile, the chosen catalysts are relatively simple, I mean, compared with the solid catalysts in heterogeneous catalysis. I was skeptical about the extension application of this code in reactions on the surfaces of solid materials, which are clearly better options for test purposes. Other well-developed codes such as RMG (Angew. Chem. Int. Ed. 2023, 62, e2023065) and Catkit (J. Phys. Chem. A 2019, 123, 2281–2285) also worked very well for automation and catalyst design.

- **1-A1:** We have extended the manuscript with a fourth example of a heterogeneous catalyst that has been studied with other automated approaches. This also includes a discussion of various state-of-the-art approaches for the automated exploration of heterogeneous catalysis, including the mentioned articles by the reviewer.

Comments of Reviewer 2 and 3

The article “Navigating chemical reaction space with a steering wheel” describes the capabilities of the STEERING WHEEL algorithm for human-guided automated exploration of complex chemical reaction networks via quantum chemical simulations. STEERING WHEEL provides an intuitive framework for human operators to define the exploration protocol and allows to decide on future steps based on the outcome of previous steps. Overall, I think the paper is well-written, describes a significant advance of the state of the art in the field and is of broad interest in chemistry. However, I think there are still multiple issues that require additional attention by the authors before publication. Hence, I recommend major revisions before this work can be published. Please find my detailed comments below.

Major issues:

- **2-Q1:** One part that I think is currently missing in the main text is a brief explanation of all the possible network expansion steps for the steering protocols. The authors provide steering protocols multiple times in the text but it is unclear what they actually constitute. This is somewhat explained in section 5.3 but I think summarizing the key features of these steps in a table in the main text would be crucial. Additionally, it is not clear how the authors came up with these protocols. Only the following is mentioned in the text: "This steering protocol is assembled in terms of keywords – such as 'Dissociation' to initiate the search for specific dissociation reactions – by a human operator on the fly." So I imagine that based on the outcome of a previous step, a human operator decides on what next step to choose. I think it would be insightful to outline how the final steering protocols were arrived at and not only have the final protocol there. I understand the final protocol is important for reproducibility but I think the process of building the protocol is as important if the reader wants to apply the tool to their specific problem. This is addressed somewhat in the Ziegler-Natta case study but I think this should be elaborated on more extensively also in the propylene hydrogenation case study and the Monsanto process case study. I am particularly interested to read how a specific observation makes the reader choose one over the other step.
- **2-A1:** We agree with the reviewer, that we have not discussed the existing implementations and practical aspects of constructing the exploration protocol in sufficient detail. Therefore, we have added section 5.4 in which we present these aspects thoroughly (including the recommended tables).
- **2-Q2:** Similarly, I am missing a more technical description of what the implemented network expansion and selection steps actually do. In my opinion, currently, this is only described in a relatively superficial manner in the methods section of the manuscript.
- **2-A2:** This is covered in the new section 5.4 as well.
- **2-Q3:** The paper presents case studies showing the application of STEERING WHEEL to study catalytic mechanisms of transition metal-catalyzed reactions. What I am missing are comparisons to classical expert-guided computational explorations of the same mechanisms. For instance, here are two references that could be compared to for the first case study (there might be others): <https://doi.org/10.1021/ja00220a010>, <https://doi.org/10.1246/bcsj.20120113>
- **2-A3:** We have added these references to the manuscript. However, we emphasize that it is not the focus of this paper to discuss the mechanistic details of all systems across the literature of the last 40 years. This would be far beyond the scope of our work. We have mainly focused our discussion on what is discussed in the reviews cited in our work. We have adapted our manuscript to clarify this aspect.

- **2-Q4:** Page 11: The authors state the following: "Also for intermediate w3 we found cis- and trans-isomers. The only two possible intermediate configurations that were not found in our exploration were the two cis-phosphine conformers with either H or Cl in trans position to the η^2 -bound propylene." Subsequently, the authors speculate that these intermediates are likely not accessible. I think it would be best to simply check this by computing these intermediates by hand and confirm that they are inaccessible.
- **2-A4:** We have constructed a structure with the propylene ligand trans to a hydrogen atom in a cis-phosphine structure. Upon structure optimization, the propylene ligand is moved further away from the ruthenium ion. The structure can be considered a stable encounter complex with the distance between the terminal carbon nucleus in propylene to the ruthenium ion being 2.6 Å with a Mayer bond order of 0.22. This is not considered a covalent bond in our framework, which is why this was not considered a successful association reaction of the propylene molecule with the catalyst in the automated exploration. Given that a full association of the propylene molecule is thermodynamically disfavored, as shown by the optimization, and that this potential reaction competes with the association reactions leading to other energetically favored stereoisomers, we deem this unsuccessful reaction trial as correct and have not carried out further reaction explorations from that stereoisomer.
- **2-Q5:** Page 12: The authors state the following: "By contrast, neural network potentials are as fast to evaluate as classical force fields, but achieve the accuracy of the reference data (typically DFT)." I think this statement needs to be supported by literature. It seems to me that the authors refer here to system-specific neural network potentials. I think this needs to be worded more clearly. This is important as I think most general neural network potentials are slower than classical force fields and largely comparable to semiempirical quantum chemistry methods. This has been discussed extensively in a recent review: <https://doi.org/10.1021/acs.chemrev.0c01111>
- **2-A5:** This is true, but it does not compromise our argument in principle, because our exploration framework works well with 'semi-empirical speed' provided that the approach is reliable in terms of stationary-point prediction. We have now adapted the text accordingly and included further references.
- **2-Q6:** Page 15: The authors state the following: "The reason for this challenge is that one cannot define a set of graph-based rules that evaluate all intermediates of the reaction mechanism given in Fig. 7 as reactive while also restricting the search space in such a way that a combinatorial explosion of reaction trials is avoided." I do not understand how the authors can state that it is impossible to define a set of graph-based rules. I think the authors should elaborate on that.
- **2-A6:** This statement tries to convey that the chemical space spanned by graph rules that allow the exploration of both the hydrolysis / condensation reactions

(inner circle in Fig. 7) and the organometallic reactions (outer circle in Fig. 7) at all times during the explorations is too large to be sampled with single-ended methods. This is solved by the STEERING WHEEL because the reactive species and reaction types can be changed in the course of the exploration. In comparison, if we were to apply all applied expansion steps on the super-set of all selection steps on the complete reaction network (which can be seen as a close equivalent to having the same reactivity rules and reaction types during the complete exploration), we would need to carry out 50,000,000 on top of the 47,000 carried out reaction trials. We understand that this might not have been clear in the original text and have rephrased the cited paragraph.

- **2-Q7:** Page 16: The authors state the following: "This search yielded the expected catalytic path, schematically shown in Fig. 8 (A), but also a stoichiometric reaction of methanol to acetic acid that consumes the catalyst by forming compound m9 shown in Fig. 8 (B). "What does 'consumes' here mean exactly? Initially, my interpretation was that this is a catalyst deactivation process. However, looking at the structure of m9, I would expect it to be possible to react with another molecule of CO to eventually get back to intermediate m5. Did the authors identify such a pathway? If this is indeed a facile process, I would not use the word 'consumes' but I would rather call it an alternative mechanistic pathway. At least thermodynamically, it does not seem to be an energy sink so I would imagine that one can get back to catalytically active intermediates.
- **2-A7:** The reviewer's observation is correct, the found reaction path could be catalytic if the final rhodium species is reacted with carbon monoxide to regenerate a previous intermediate (**m5**) of the reaction path. Before the initial submission, we had searched the reaction network for such a reaction, but it could not be found. However, this can be attributed to the fact that we have not carried out reaction trials for such a step, which we noticed during the revision. We have now added an **Association** step as suggested by the reviewer. This step carried out 15 further reaction trials and found the association reaction which closes this alternative catalytic cycle. We have adapted the results surrounding Fig. 8, concluding that the new path is indeed an alternative catalytic path, and have updated the statistics in Tab. 1 in Section 5.3.
- **2-Q8:** Page 19: The authors state the following: 'Since Feliz et al. observed a strong effect of the electronic structure model and solvent description on the initial transition state [120], we suspect that this elementary step is highly disfavored in the tight binding method that we relied on for the initial structure exploration.' As in one of my comments above, I think it would be good to simply calculate this step explicitly to check whether the hypothesis that this elementary step is simply highly unfavorable at the level of theory used is correct.
- **2-A8:** Before submission, we had already carried out manual searches for the transition state of this reaction and could not locate it with GFN2-xTB. We have added

this information to the main text. We now report on an exploration where the first 3 expansion steps in our protocol were carried out with a changed underlying electronic structure model (PBE-D3/def2-SVP(CPCM)). We have adapted the third selection step to be more restrictive in order to carry out less reaction trials because of the increased computational costs per energy evaluation. Indeed, this repetition of our protocol with a changed model yielded the expected S_N2 mechanism with the rhodium ion as the nucleophile and iodide as a leaving group. In order to complete the two-step process we had to add a fourth expansion step, which probed association reactions of iodide with rhodium catalysts. We have adapted the results section in our manuscript accordingly, and we have uploaded the reaction network based on DFT as a separate database on Zenodo.

Minor issues:

- 2-Q9: Figure 6: I think the radical dots should be made a bit thicker, I find them easy to be overlooked.
- 2-A9: We have increased their size.
- 2-Q10: The authors state the following: "This path is therefore an excellent example to demonstrate on how Chemoton can uncover new reaction mechanisms, enhanced by the intuitive reaction network analysis with Pathfinder in Heron that allows one to export a graph as depicted in Fig. 8, simply by pressing a button in the graphical user interface." I find this statement slightly misleading. Initially, I was quite impressed with this statement. However, when reading the caption of Figure 8, the authors state: "The two path diagrams A and B were directly exported from Heron and then manually augmented with Lewis structures." Hence, I think the authors should already mention in the main text that the figure is augmented. In that regard, I wonder whether the non-Lewis structures (chemical sum formulas) were also added later or whether they are added by the program. Additionally, are the arrows and compound labels also added by hand?
- 2-A10: We have adapted the main text concerning Fig. 8 to clarify which parts of the figure we augmented manually after the export. In its current form, the energy diagram can be exported with or without the chemical sum formulas of each intermediate. These sum formulas do not include chemical heuristics, *e.g.* methanol would be written as CH_4O instead of H_3COH . Because our graphical user interface provides interactive three-dimensional depictions of each intermediate and spline fits to the intrinsic reaction coordinate of each transition state and because the plot can be exported in the SVG format, we find the augmentation with Lewis structures or additional information to be straightforward. For illustration, this would be the figure directly exported from Heron:

- 2-Q11: This is not really an issue but more of a comment. The authors state the following: "In addition, we found a single-step process, in which the methyl-iodide bond was concertedly broken and the methyl group was bound to an existing CO ligand instead of to the rhodium center, directly forming intermediate m3a, which is a stereoisomer of the compound m3, shown in Fig. 7." Personally, I would refer to this step as a pericyclic reaction as it has a cyclic transition step and is concerted. I would even be tempted to refer to it as a cycloaddition, even though, strictly, cycloadditions involve "unsaturated molecules".
- 2-A11: We thank the reviewer for the comment and have adapted the main text. The reaction is in accordance with the definition of a pericyclic reaction.

Comments of Reviewer 4

The authors introduced a steering wheel algorithm, which provides an intuitive means of directing the exploration of reaction networks. This algorithm incorporates both a network expansion step and a selection step, enabling the generation of conformers to expand chemical reaction networks while also constraining the explored chemical space. To illustrate its effectiveness, the authors have applied this algorithm to three widely recognized transition metal catalysts, yielding promising results. I believe that this algorithm holds the potential to efficiently automated reaction mechanisms in a more controlled way. In order to recommend publication, the following concerns need to be addressed.

- 4-Q1: In Section 2, while the authors attempt to present the conceptual design and implementation of the steering wheel, it is very hard to follow without prior knowledge of SCINE and CHEMOTON. The authors should provide a more detailed explanation of the connections between these elements and clarify the logical integration of the steering wheel.
- 4-A1: We deliberately kept this section separate from the existing SCINE framework as the described issues and suggested solutions are generally applicable and

could be transferred to other automated reaction network exploration software. However, we understand that this might make the paper more difficult to read and have therefore introduced a new paragraph in section 2 that explains the larger framework into which our new approach is implemented with the intend to be understandable for people unfamiliar with our work. Together with the newly added technical description of our implementations in section 5.4, we think that our approach can be much better understood.

- **4-Q2:** The authors have provided three case studies. Nevertheless, in order to more effectively showcase the benefits of the steering wheel concept, it is advisable to include a fair comparison with previous approaches. This comparison could encompass factors such as the number of calculations and suggested intermediates. If only the steering wheel method proves effective in these case studies, the necessity of this approach should be underscored.
- **4-A2:** We agree with the reviewer that a proper comparison of different automated exploration frameworks with respect to the Steering Wheel concept is desirable. However, we would like to emphasize that this is an open problem in general and has not been accomplished by anyone working in the field (for various reasons). As an example, we may refer to the multiple challenges encountered in the study of Green *et al.* (10.1021/jacs.7b11009) and the response by Maeda *et al.* (10.1021/acs.jctc.8b01182). These two papers shows the general difficulties in such comparisons. Future work should solely focus on this issue, but it would be far beyond the scope of the present paper.
However, in view of the reviewer’s comment and the feedback from reviewer 1, who wondered whether our approach also works for heterogeneous systems, we have selected a system which has already been studied with another automated approach and have compared the mentioned metrics to the best of our knowledge (see comment above).
- **4-Q3:** In these three case studies, GFN2xTB were employed. However, structure refinement through DFT was only applied on Wilkinson catalyst (first case study). Since tight binding approach may lead to wrong structure, this omission may introduce inaccuracies into the resulting reaction mechanisms and proposed intermediates for the second and third case studies. The authors should address how this choice impacts their findings.
- **4-A3:** We have addressed this for the Monsanto process on Page 20 for which we could not recover a known S_N2 elementary step that was already found to be dependent on the employed density functional and implicit solvation model. We have studied this aspect in more detail in the revision and have added results to the Monsanto reaction network for which we have carried out our exploration protocol with a more reliable electronic structure model and could recover the known mechanism, which we see as a strong point for the generalizability of our approach to multiple electronic structure methods and the systematic improvability

by repeating reaction trials for crucial steps in the network with more accurate methods.

We also stress that our paper presents a new methodology rather than a final study of specific reaction mechanisms (for the latter, we would consider further exploration and subsequent validation by coupled cluster calculations; which can be activated in SCINE, but which would make each reaction mechanism a subject in its own right, worth writing separate papers). We have emphasized the effect of uncertainties on our current data in the Conclusions.

- **4-Q4:** In each case study, the authors have provided a list of network expansion steps, exemplified by [Dissociation, Association, Association, Rearrangement, Rearrangement, Dissociation] in the case of Wilkinson catalyst. It is my belief that the sequence in which these expansion steps are undertaken can significantly influence the resulting reaction mechanism. Therefore, the authors should clarify how one can systematically consider the order of expansion steps without making prior assumptions about the reaction mechanism, and how to handle the case that multiple steps happen simultaneously.
- **4-A4:** In fact, this is not a bug, but a feature. The default setting of our exploration methodology is an unbiased brute-force exploration, but such an approach is truly time-consuming and does not deliver chemical insights quickly. The STEERING WHEEL in its current form with a human in the loop builds on the fact that there usually exists one or more hypotheses about a reaction mechanism prior to its exploration. In some sense, the STEERING WHEEL can be seen as an algorithmic interface to encode mechanistic hypotheses. In all four case studies, the exploration protocol was designed in line with mechanistic ideas from the literature. However, we could show that, first, we can discover intermediates and reaction paths that were not directly assumed but are close to the proposed mechanism, and second, that a failure in our original assumption can be corrected on the fly, as in our newly added example in section 3.4, where the mechanism presented in the literature included a mistake of labelling different carbon nuclei in a figure incorrectly. After the expansion step based on the literature failed to advance the reaction network, we had looked at the mechanism in more detail, could spot the mistake, and then add the expected reactivity to the exploration, which proved to be successful. In the same fashion, an original proposal of an otherwise unknown reaction mechanisms can be adapted during the exploration process, if the exploration fails to find hypothesized intermediates or finds that they are only accessible via a transition state too high in energy. We also want to stress that our exploration can be expanded at any point to full exhaustive explorations, but the new feature is that we can adjust this degree precisely.

The reviewer’s remark about multiple steps that could happen simultaneously is valid and has already been considered in our original design. Such a scenario is solved in our framework by either choosing an expansion step that can cover all competing steps or by covering a part of the potential reactions with one expansion

and then adding a selection that specifies that the last selection step should be repeated. Then, on this previous selection another expansion can be applied to in such a way that all possible simultaneous steps are covered. This allows for something similar to a branching exploration protocol, while not violating our linear constraint, which is necessary for reproducibility. Even if competing elementary steps were first not considered, any expansion step can work on the whole reaction network, meaning that any part of the mechanism can be studied in more detail later on with additional calculations. We thank the reviewer for their careful evaluation and have now included the discussion of such cases in the manuscript.

- 4-Q5: In alignment with point 4, since the definitions of expansion steps and selection steps are crucial to the steering wheel algorithm, I recommend that the authors provide a more comprehensive explanation, supplemented by illustrative examples of their design. This would not only enhance comprehension but also facilitate a better understanding of the steering wheel interface (as shown in Figure 2).
- 4-A5: Because this was a request also by other reviewers, we paid special attention to this aspect while writing section 5.4 (see above).

Further changes to the manuscript

- Adapted sections to publishing guidelines.
- The citation to the AutoRXN workflow was corrected.
- The script that generated figure 8 did not account for the case that reactions can be stored in their reversed direction in our database, which was the case for a single reaction per path. We deposited the corrected script on Zenodo and updated the figure. The conclusions drawn from the figure are not affected.
- The statistics of the Wilkinson network in Table 1 were off due to a mistake in a database query, the numbers were corrected.

REVIEWERS' COMMENTS

Reviewer #1 (Remarks to the Author):

My concerns in the first round of review have been well addressed. The revised manuscript can be recommended to be accepted.

Reviewer #2 (Remarks to the Author):

The article "Navigating chemical reaction space with a steering wheel" is the revised version of a manuscript that I reviewed previously. I think the authors did an excellent job in addressing the comments of the reviewers and improving the manuscript. Especially the addition of the case study on the silica-supported single-site catalyst is really impressive and demonstrates the power of the approach. Hence, it is my pleasure to recommend this article for publication. I have only one extremely minor comment about one of the implemented changes that I think should be addressed.

The one thing I do want to mention is that "concerted pericyclic reaction" (page 21) is a pleonasm and should simple be "pericyclic reaction" in my opinion.

Reviewer #3 (Remarks to the Author):

The authors have sufficiently addressed the comments and concerns and the revised version is suitable for publication.

Reviewer #4 (Remarks to the Author):

In my view the reviewer comments have been adequately addressed in the revision. I therefore recommend this paper for publication.

List of Changes for Manuscript NCOMMS-23-40908

“Navigating chemical reaction space with a steering wheel”

Miguel Steiner and Markus Reiher

We thank the reviewers for their time and effort and for their recommendation to publish the revised manuscript. In the following, we address the remaining issue raised by one reviewer.

Comments of Reviewer 2

The article “Navigating chemical reaction space with a steering wheel” is the revised version of a manuscript that I reviewed previously. I think the authors did an excellent job in addressing the comments of the reviewers and improving the manuscript. Especially the addition of the case study on the silica-supported single-site catalyst is really impressive and demonstrates the power of the approach. Hence, it is my pleasure to recommend this article for publication. I have only one extremely minor comment about one of the implemented changes that I think should be addressed.

The one thing I do want to mention is that ‘concerted pericyclic reaction’ (page 21) is a pleonasm and should simple be ‘pericyclic reaction’ in my opinion.

- *We agree with the reviewer and removed ‘concerted’ in the mentioned sentence.*